# Strategy Trade-Off of Predominant Stress Tolerance Relative to Competition and Reproduction Associated with Plant Functional Traits under Karst Forests

Xiaorun Hu [1], Yuejun He [1,*], Lu Gao [1], Muhammad Umer [1], Yun Guo [1,2], Qiyu Tan [1], Liling Kang [1], Zhengyuan Fang [1], Kaiping Shen [1] and Tingting Xia [1]

[1] Forestry College, Research Center of Forest Ecology, Guizhou University, Guiyang 550025, China; hxr7013@163.com (X.H.); gl19970802@163.com (L.G.); umer@gzu.edu.cn (M.U.); zihanyun2013@163.com (Y.G.); tqy0820@163.com (Q.T.); kangliling0727@163.com (L.K.); 15761636506@163.com (Z.F.); skp0825@163.com (K.S.); xtt1268@163.com (T.X.)

[2] College of Eco-Environmental Engineering, Guizhou Minzu University, Guiyang 550025, China

* Correspondence: hyj1358@163.com

**Abstract:** The Grime (1974) CSR framework posits that ecological strategies of competition, stress tolerance and ruderal reflect plants' adaptability to their survival environments. Karst forests are crucial for terrestrial ecosystem functions. However, how karst forests regulate plant functional traits in ecological strategy to adapt to infertile habitats remains unclear. Therefore, we surveyed fifty-three karst forest plots and measured plant functional traits involving the plan diameter at breast height (DBH), height, leaf area (LA), specific leaf area (SLA), leaf dry matter content (LDMC), leaf carbon (LC) and nitrogen (LN) and phosphorus (LP) with the leaf water content (LWC). We calculated CSR components on the individual and community levels introduced by community-weighted means (CWM) using the 'StrateFy' calculator. Principal component analysis (PCA) and Mantel's test were used to investigate trait correlations with CSR components. Our results showed that stress tolerance (S) contributed an average 65.88% and 63.63% in individuals and communities, respectively, followed by competition (C) at 25.82% and 29.63%, and the least, ruderal (R), at 8.30% and 6.74%. Different plant functional traits exhibited different variations, coupled with significant correlations between CSR components and PC1 scores (except for CWM− LA, SLA and LDMC). Component S increased with the increase in CWM− LC and C:N ratio, and decreased with the increase in CWM− DBH, Height, LWC, LN, and N:P ratio, while it was the opposite for C and R, highlighting strategic trade-offs associated with plant functional traits. Mantel's test revealed varied key trait combinations for each strategy. In conclusion, the predominant stress tolerance strategy relative to competition and ruderal is a result of trade-offs regulating karst forests, in association with plant functional traits. The disentangled CSR strategies provide insights into theoretically understanding functional maintenance for infertile forest ecosystems as an evolutional regulation mechanism.

**Keywords:** CSR theory; ecological strategy; trade-off; community-weighted mean (CWM); plant size; leaf macronutrients

## 1. Introduction

Grime (1974) raised the CSR framework regarding ecological strategy on competition (C), stress-tolerance (S), and ruderal (R) for organisms [1]. Specifically, strategy C is mainly to improve the competitive ability, strategy S is mainly to grow maintenance under limited resource conditions, and strategy R is mainly to cycle fast for life reproduction in suffering frequent interference [2,3]. CSR theory can not only be adopted to explain the ecological process of community integration, biodiversity maintenance, and vegetation succession [4–6], but also the ecosystem function productivity and community stability [7,8]. Plant survival

could be reflected by ecological strategies via plant functional traits responding to environmental changes [9,10]. Therefore, it is necessary to clarify ecological strategies through plant functional traits for explaining vegetation responses to environmental variations.

Traits of plants are plant-measurable characteristics in evolutional adaptation, including the external morphology and the internal physiology of leaves in responding to environments [11]. Leaf functional traits determine adaptability in the strategic trade-offs for plants to environments [9,12]. The leaf economics spectrum (LES) is functional combination of a series of balanced or synergistically varying functional traits that can quantify plant resource trade-off strategies [13]. On the one hand, leaf traits of leaf area (LA), specific leaf area (SLA), and leaf dry matter content (LDMC) in phenotype [14], and the macronutrient C, N and P support light capture ability due to the material basis [15], while their stoichiometric ratio reflects leaf structure and nutritional efficiency [16]. On the other hand, plant size includes plant diameter at breast height (DBH) and plant height [17]. Previous studies have shown a significant correlation between leaf functional traits and plant size [18]. Plant size may significantly affect plant physiological traits, especially those related to plant photosynthetic capacity and nutrition [19], determining the survival, growth and reproduction for plant species. According to previous studies, plants trade off their growth, survival, and reproduction to adapt to resource conditions by regulating plant functional traits [20–22]. However, due to the differential individual and community scales, single plant traits in physiology and morphology cannot predict the eco-processes of a community [23].

The community-weight mean (CWM) of traits reflects the community characteristics and can reveal the adaptive strategy at the community level [24,25]. Meanwhile, CWM traits of plants are affected by environmental conditions such as soil nutrients, topography, climate changes, etc., supplementary regulation changes of CSR strategy [26,27], especially in harsh ecosystems generally suffering the scarcity of nutrients or water. Consequently, CWM traits were used to evaluate ecological strategies of CSR at community levels, which can better describe the plant community response to environmental changes [28,29].

The karst landform in southwest China, a fragmental ecosystem, is the most extensive-continuous distribution representing typical development [30,31]. Landscape fragmentation of karst leads to different microhabitats, such as stone caverns, grooves, surfaces, and soil surfaces [32]. These microhabitats are interlaced distribution each other, resulting in a highly heterogeneous habitat [33,34], due to high rock exposure rate, discontinuous soil cover, and poor water retention led to infertile soil such as N or P [4,35]. Further, these heterogeneous habitats affect plant composition and diversity, also regulating plant functional traits at individual and community levels [36,37]. Generally, karst forests developing high LDMC, LC, and low SLA, LN [25], maintained high species diversity [38]. Liu (2012) and Wang (2022) found that the karst community showed a conservative strategy compared to a non-karst community [39,40], showing an acquisitive strategy under the same climate regulation [41]. However, how karst forests strategically regulate plant functional traits adapting to infertile habitats remains unclear at present.

Most species exhibit high-stress tolerance in the degraded ecosystem by adopting the S strategy [42]. Rios (2022) showed that S strategy species can better adapt to harsh environments, followed by C strategy [43]. Therefore, we hypothesized that most karst woody plants and communities adopt the predominant S strategy followed by C and R (H1). Plant functional traits are important attributes in characterizing plant adaptation to the environment in functional evolution [44]. Plant functional traits determine adaptability in the strategic trade-offs for plants to environments [9,12]. Hence, ecological strategy trade-offs are associated with plant functional traits in regulation through long-term evolution adaptivity [45–47]. Therefore, we hypothesized that CSR strategy depends on trade-offs in regulations associated with plant functional traits in karst forests (H2). The aim is to explore ecological strategy regarding competition, stress tolerance and ruderal based on the CSR framework association with leaf functional traits for understanding functional regulation mechanisms in karst forest ecosystems.

## 2. Materials and Methods

### 2.1. Field Community Survey

A field survey was conducted in a typical karst forest located in Machang, Xiayun, and Gaofeng Town, Pingba District of Anshun city, Guizhou Province, China (E: 105°59′24″~106°34′06″, N: 26°15′18″~26°37′45″, altitude: 963 m~1645.6 m; Figure 1). It belongs to the subtropical monsoon climate with annual mean precipitation 1200 mm and annual average temperature 13.3 °C. The vegetation is a mixed evergreen and deciduous broad-leaved forest, indicating typical non-zonal vegetation. A total of 53 plots (20 m × 20 m) were surveyed in containing total 212 subplots of 10 m × 10 m, randomly through selecting plot in this area and approximately average distance 2 km when the minimum and maximum distance 50 m and 4.5 km, respectively, among plots. We identified the plant species composition with its number, height (m), and diameter at breast height (DBH, cm) of plants with DBH ≥ 1 cm in height more than 1.3 m. In particular, the surveyed forest community could be divided into three vertical layers of trees, shrubs, and herbs according to the plant height and species property. In the factual survey process, those woody plants with heights ≥ 5 m were recorded into the tree layer dominated by *Itea yunnanensis*, *Platycarya strobilacea*, *Carpinus pubescens*, *Quercus phillyraeoides* and *Ligustrum lucidum*, otherwise as the shrub layer dominated by *Myrsine africana*, *Carpinus pubescens*, *Rosa cymose*, and *Coriaria nepalensis*, while herbs were a separate layer under the canopy as the dominant plants including *Cyrtomium fortune*, *Setaria faberii*, *Arthraxon hispidus*, *Ficus tikoua*, etc. Additionally, we collected healthy branches of woody plants through four orientations in the surveyed plots. All branches were taken and kept in a freshness protection packaging for further measuring leaf functional traits.

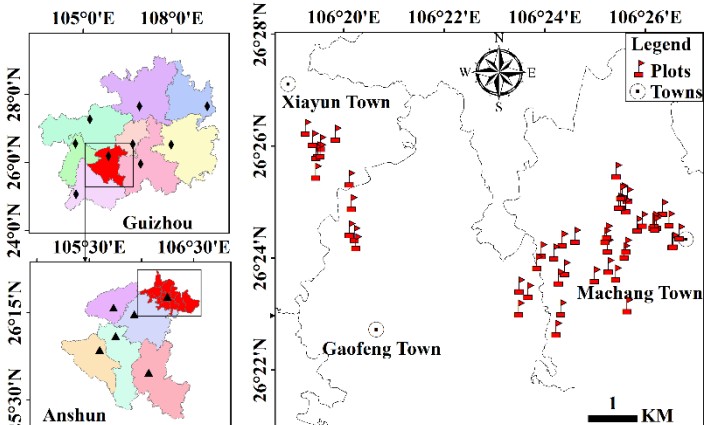

**Figure 1.** Field plots location and distribution. The black diamond shape represents different cities in Guizhou Province, and the black triangle represents different districts of Anshun City.

### 2.2. Plant Trait Determinations and Community-Weighted Mean

Leaf traits of 92 woody species out of 1305 individuals from 53 plots were measured using the method of Cornelissen [14]. In detail, the leaf area (LA, cm$^2$) used a foliar digital scanning system (WinRHZIO_Pro LA2400); the leaf-fresh weight (LFW, mg) was weighted by an electronic balance in accuracy 1/10,000; the leaves were rehydrated for 6–12 h before measuring the leaf dry weight (LDW, mg), then leaves were oven-dried at 65 °C for 48 h to constant weight and weighted as LDW. In particular, the specific leaf area (SLA, cm$^2$/g) was LA divided by LDW; the leaf dry mass content (LDMC, %) was LDW divided by LFW; the leaf water content (LWC, %) was the difference between LFW and LDW divided by LFW. In addition, the leaf carbon, nitrogen, and phosphorus content (LC; LN; LP; mg/g) in dried-weight were determined using the potassium dichromate, Kjeldahl, and molybdenum antimony [48]. In turn, we calculated the leaf-stoichiometric ratios of C:N, C:P, and N:P. In total, it referred to items of 14,870 for each leaf trait from those woody individuals of DBH ≥ 1 cm in height more than 1.3 m in all plots. Subsequently,

the individual trait mean was calculated for each individual to explore the CSR strategy. The CWM of community-weighed mean on leaf traits can reflect community functional traits [24]. Thus, we used the CWM value to explore the CSR strategy of communities in this study. According to Garnier [49], the CWM value of each trait can be obtained as follows:

$$\text{CWM} = \sum_{i=1}^{n} P_i \times \text{Trait}_i$$

where CWM is the plant functional traits at the community scale; $n$ is the species numbers in a plot; $P_i$ is the relative abundance of species $i$ in a plot; and $\text{Trait}_i$ is the trait value of species $i$.

### 2.3. CSR Component Calculations

Based on the CSR scheme from Grime [1], a global CSR strategy calculator of 'StrateFy' was established by Pierce using trait variation of LA, SLA, and LDMC in vascular plants to evaluate CSR strategies of plants [12]. The method has been reliable in locating the CSR strategy for plants [50,51], and it was also adopted widely to evaluate the strategic difference of CSR for plant communities in various regions [52,53]. Therefore, we calculated the component values C, S, and R for each species integrating community through the spreadsheet of 'StrateFy' provided by Pierce at https://besjournals.onlinelibrary.wiley.com/doi/full/10.1111/1365-2435.12722 [12], accessed on 10 April 2022. In detail, CSR component values of individuals in 'StrateFy' tool using individual trait mean of LA, LDMC, and SLA from 1305 individuals of 53 plots. Likewise, based on the CWM values of LA, LDMC, and SLA were subjected to 'StrateFy' analysis and obtained the component values of CSR in each plot. Then, the CSR component values of individuals and community were created the ternary plots by 'Ternaryplot' function in R software (v.4.1.3, R Core Team, Vienna, Austria) [54].

### 2.4. Data Analysis

Plant functional trait variations were analyzed using SPSS software (v.24.0, New York, NY, USA). We took the principal component analysis (PCA) to explore associations among plant traits at the community level and drew graphs by Origin 2021 (OriginLab, Northampton, MA, USA). Meanwhile, we used PCA scores excluded LA, SLA and LDMC to analyze the correlation with CSR components. In addition, the association between the plant functional traits and strategic components was analyzed, and regression graphs were drawn by Origin 2021 (OriginLab, Northampton, MA, USA). Further, Mantel's test was performed to disentangle the key trait combinations in strategy regulation by 'linkET' package in R software (v.4.1.3, R Core Team, Vienna, Austria) according to Sunagawa [55].

## 3. Result

### 3.1. CSR Strategies for Individuals and Communities in the Karst Forest

It was evident from the analysis of the component distribution of 1305 woody individuals, obtained from traits of LA, SLA and LDMC by the calculator 'StrateFy', that they mainly clustered on the C-S side favoring the S side (Figure 2A). Component S had an average of 65.88%, ranging from 20% to 97%, while component C had an average of 25.82%, ranging from 3% to 60%. Additionally, component R reached an average of 8.30%, ranging from zero to 50%. In addition, the CWM traits of LA, SLA and LDMC of 53 plots were also located on the C-S side of the community CSR ternary plot (Figure 2B). Here, component S had an average of 63.63%, ranging from 45% to 82%, and component C had an average of 29.63%, ranging from 18% to 40%. Component R had an average of 6.74%, ranging from zero to 15% in the respective plots of the surveyed region.

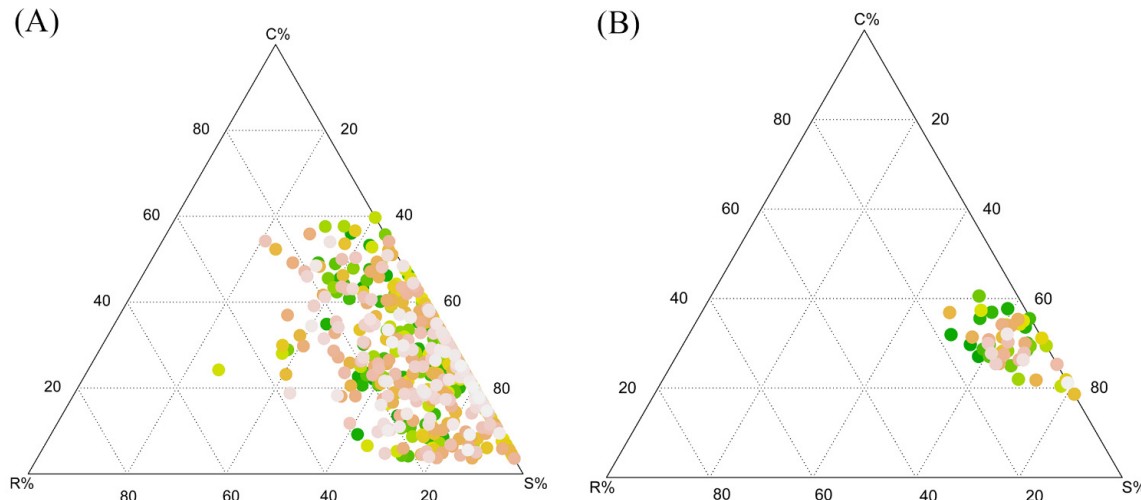

**Figure 2.** Ternary plot showing the strategic components of CSR regarding the karst individual (**A**) and community (**B**). The C%, S%, and R% arranged at the triangle vertex indicate the percentage of C, S, and R components by 'StrateFy' analysis, respectively. The different color circles represent different individual or community strategic value.

### 3.2. Variation and Differentiation of Plant Functional Traits of Karst Forests

The coefficients of variation (CV) for plant functional traits ranged from 2.9% to 39% (Table 1) by the greater variation of LA (38.77%), DBH (32.45%) and height (31.87%) and the lower variation of LC (2.96%), LWC (5.48%) and LDMC (6.58%) in community-weighted means. In addition, PCA analysis showed the relationship among plant functional traits (Figure 3). The first PCA analysis showed the variations of all plant functional traits were explained by 39.9% at PC1 and 26.7% at PC2, respectively (Figure 3A). Additionally, the trait differentiation by the CWM− C:N ratio and LC and LDMC as the conservative resource strategy presented at the −PC1 axis, but other traits presented the +PC1 axis as the resource acquisition strategy, evidently reflecting the leaf economic spectrum at the PC1 axis; also, the first PCA showed negative correlations between CWM− SLA, LA, LWC, LN, DBH, Height and CWM− LDMC, LC (Figure 3A). Moreover, the second PCA showed that the variations of plant functional traits excluding CWM− LA, SLA, and LDMC were explained by 36.6% at PC1 and 35.1% at PC2, respectively (Figure 3B); meanwhile, the trait differentiation by the CWM− C:N ratio and LC as the conservative resource strategy presented at the −PC1 axis, but other traits presenting the +PC1 axis as the resource acquisition strategy. Similarly, the PCA also showed negative correlations between CWM− LA, LWC, LN, DBH, height, CWM− LC, and C:N ratio (Figure 3B).

**Table 1.** The variation of plant functional traits in plant size and leaf traits.

| Plant Traits | Mean ± SD | Minimum | Maximum | CV (%) |
|---|---|---|---|---|
| LA (cm$^2$) | 25.78 ± 9.99 | 8.14 | 56.58 | 38.77 |
| SLA (cm$^2$/g) | 123.74 ± 22.30 | 77.88 | 191.97 | 18.02 |
| LDMC (%) | 45.24 ± 2.98 | 37.04 | 54.45 | 6.58 |
| LWC (%) | 54.84 ± 3.01 | 45.68 | 63.13 | 5.48 |
| DBH (cm) | 2.65 ± 0.86 | 1.27 | 6.29 | 32.45 |
| Height (m) | 2.83 ± 0.90 | 1.20 | 6.03 | 31.87 |
| LC (mg/g) | 435.60 ± 12.88 | 399.96 | 475.15 | 2.96 |
| LN (mg/g) | 15.44 ± 1.67 | 12.57 | 22.29 | 10.78 |
| LP (mg/g) | 0.83 ± 0.20 | 0.47 | 1.79 | 24.19 |
| C:N ratio | 30.94 ± 3.40 | 20.42 | 37.34 | 10.99 |
| C:P ratio | 606.09 ± 139.60 | 311.61 | 1019.29 | 23.03 |
| N:P ratio | 20.32 ± 4.61 | 13.92 | 36.30 | 22.69 |

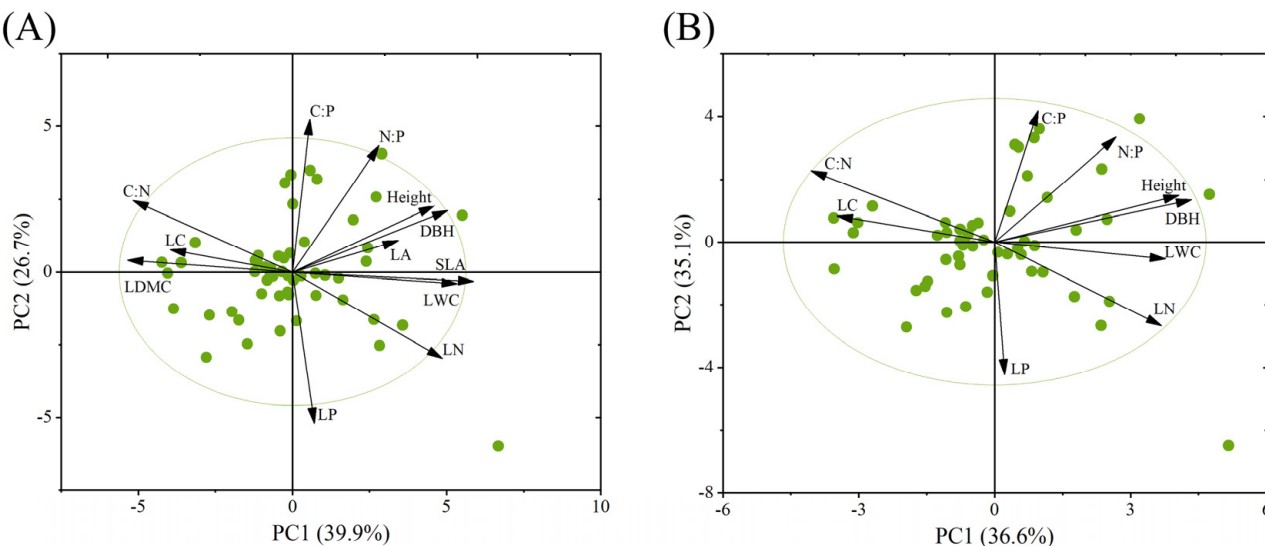

**Figure 3.** Differentiation and correlation of all traits (**A**) excluding traits (**B**) of LA and SLA and LDMC in CWM.

### 3.3. CSR Components Associated with Plant Functional Traits

As shown in Table 2, components of C and R were significantly positively correlated with the total PC1 score of traits (exclude LA, SLA and LDMC), with a significant negative correlation between component S and the PC1 score, while presenting nonsignificant correlations with the PC2 score. In addition, component S was significantly negatively correlated with CWM− DBH, Height, LWC, LN, and N:P ratio (Figure 4A–C,E,I), while being significantly positively correlated with CWM− LC and C:N ratio (Figure 4D,G), but not for CWM−LP in significance (Figure 4F). Component C was significantly positively correlated with CWM− DBH, Height and LWC (Figure 4A–C), while displaying no significant correlation with CWM− LC, LN and LP, C:N, C:P, N:P ratios, respectively (Figure 4D–I). Component R was significantly positively correlated with CWM− DBH, Height, LWC, LN, and N:P ratio (Figure 4A–C,E,I) and significantly negatively correlated with CWM− LC and C:N ratio (Figure 4D,G), while displaying no significant correlation with CWM− LP and C:P ratio (Figure 4F,H).

**Table 2.** Pearson's correlations between CSR components and PC1 and PC2 scores.

| Strategy Components | PC1 Score | PC2 Score |
|---|---|---|
| Component C | 0.440 *** | 0.110 |
| Component S | −0.810 *** | −0.018 |
| Component R | 0.790 *** | −0.140 |

The *** indicate the significant correlation at $p \leq 0.001$.

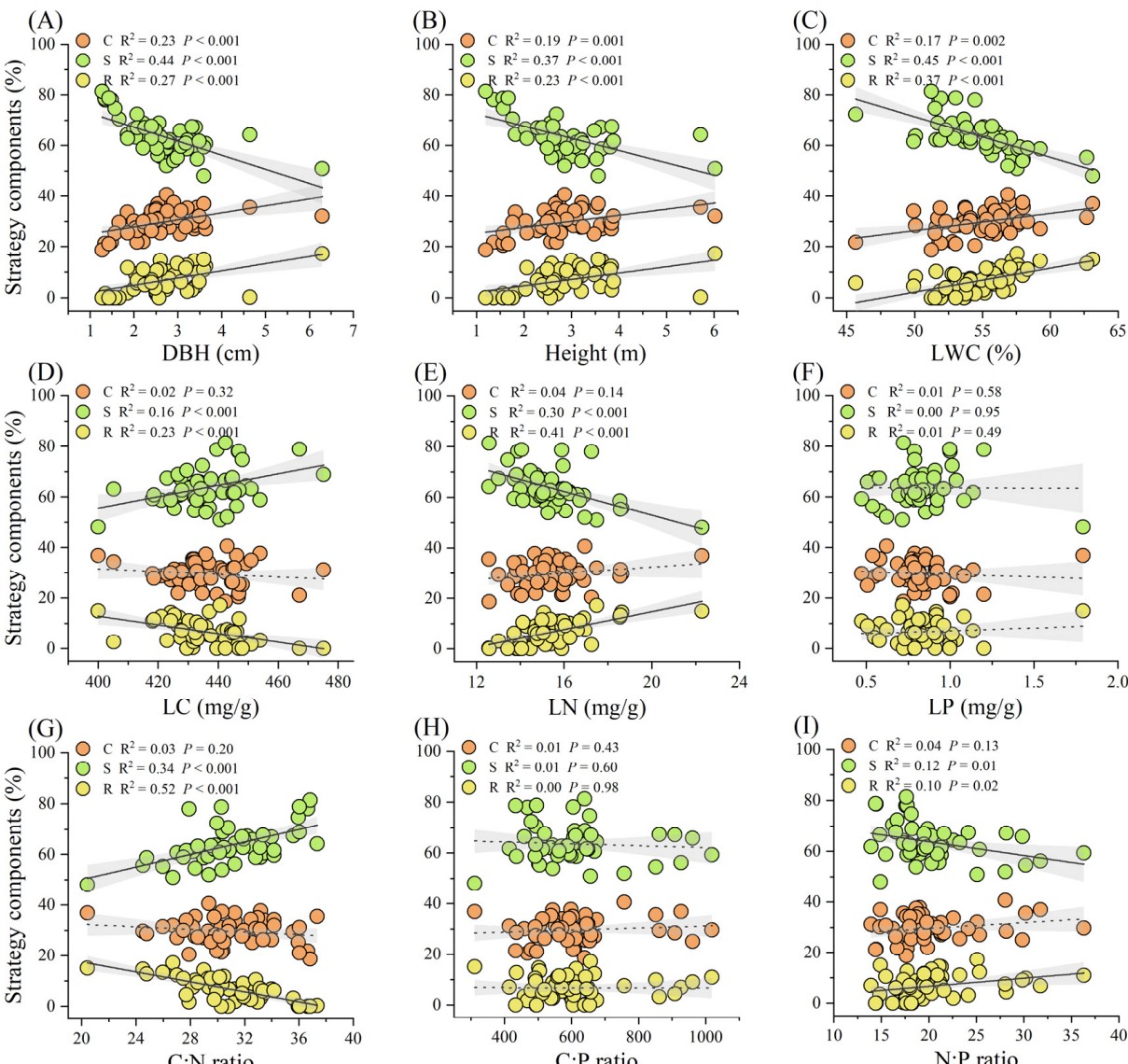

**Figure 4.** Component changes regarding CSR strategies with plant functional traits in CWM. The subgraph (**A–I**) indicate the trend of CSR components change with DBH, Height, LWC, LC, LN, LP, C:N, C:P, N:P, respectively. $R^2$, coefficient of determination; *P*, probability. The solid lines denote a significant correlation at *P* < 0.05, and the dashed black lines denote a none significant correlation at *P* > 0.05.

### 3.4. Mantel's Tests for CSR Strategies Associated with Functional Trait Combinations

Mantel's tests showed the key trait combinations associating the CSR components (Figure 5). At the individual level, component C was significantly correlated with key trait combinations of DBH, Height, LWC and LP, while component S was significantly correlated with all traits; simultaneously, component R was significantly correlated with LC, LN, LP, C:N, C:P and N:P ratios (Figure 5A). Similarly, at the community level, component C was significantly correlated with key trait combinations of CWM− DBH and Height, component S was significantly correlated with key trait combinations of CWM− DBH, Height, LWC, LC, LN, LP and C:N ratio, and component R was significantly correlated with key trait combinations of CWM− DBH, Height, LWC, LC, LN, LP, and C:N ratio except for CWM− C:P and N:P ratio (Figure 5B).

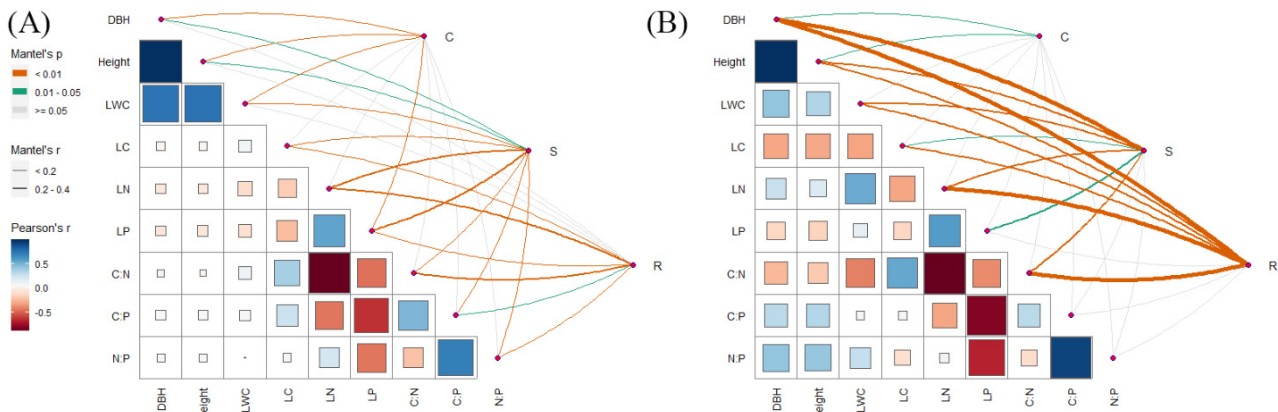

**Figure 5. Figure 5**. The Mantel's test for CSR strategy in individual (**A**) and community level (**B**). The color gradient indicates Pearson's correlation. Edge width responds to the r statistic of Mantel in the distance correlations, and edge color statistically indicates the significant level at Mantel's *p*-value.

## 4. Discussion

### 4.1. The Ecological Strategy in the Karst Forests

In this study, the strategic components mainly clustered on the C-S side to most plant individuals and communities (Figure 2), representing the predominant S strategy, followed by the C strategy and finally R. Most woody plants use the S or CS strategy, demonstrating high-stress tolerance [56]. For instance, Rosenfield (2019) and Han (2022) found that subtropical forests had a higher stress tolerance [26,52], consistent with our results of the predominant S strategy involving stress tolerance. Caccianiga (2006) suggested that the S strategy species dominated the late succession, with the decreasing dominance of the C strategy species gradually [6]. Therefore, it supported our results that the typical karst forest in the subtropical region in this study is approached to the late-successional stage of the community, thus supporting the S strategy dominance followed by the C and R strategies according to our results, which verified our hypothesis of H1. Zhou (2021) and Rios (2022) found that most species and communities exhibit high-stress tolerance by adopting the S strategy, followed by the C strategy in the poo-resource ecosystem [42,43]. According to the CSR theory of Grime (1977), the S strategy plants are mainly for survival by stress-tolerance in harsh habitat conditions, and the C strategy plants are mainly for conquest through competition due to scarce resources, while the R strategy plants are mainly for reproduction through increasing individuals of a population [2]. In particular, the karst maintains a lower resource of N and P and a barren soil layer with higher heterogeneous habitat [4,57,58], leading to adverse habitat conditions and scarce habitat resources. Therefore, competing for limited N and P resources through stress tolerance for individual survival is necessary as the optimal strategy of S with C in adverse karst habitats to maintain fewer family members. Of course, progeny reproduction is also important for karst plants. However, it is more important to prioritize the individual survival of karst plants. Our results in this study confirmed that the predominant strategy of S followed C and R was presented in subtropical karst forests.

### 4.2. Functional Trait Differentiation Associated with Plant Resource Strategy

In this study, plant traits displayed differential variations (Table 1), indicating trait differentiation companied by PCA analysis to trait correlations (Figure 3). In addition, CSR components were significantly positively or negatively correlated with PC1 scores (Table 2), which indicated that CSR strategies of the woody plant community were regulated by plant functional traits in growth size and leaf stoichiometric traits, also reflecting the plant economics strategy in resource conservation and resource acquisition. This plant economics strategy regarding resource conservation and acquisition trade-offs of leaf functional trait variation and differentiation synergistically at PCA resource axes, exactly as

shown in Figure 3, reflects the leaf economic spectrum (LES) raised by Wright (2004) [13]. For example, Wang (2023) discovered that there were strategic trade-offs and strategic differentiations of acquisition and conservation in resources through PCA analyzing leaf functional traits to both subcommunities composed of evergreen and deciduous species from karst forests [59], exactly supporting the LES theory of Wright (2004) [13]. Therefore, our results also supported the LES according to PCA analysis of Figure 3.

*4.3. Trade-Offs of CSR Strategy Associated with Plant Functional Traits in the Karst Forest*

In our results, CSR components were significantly correlated with PC1 scores (Table 2), indicating ecological strategy regulation impacted by plant functional traits, further revealing a strategy trade-off in stress tolerance relative to competition ability and reproduction through the predominant component S relative to C and R responding to community functional traits of plant size and macronutrients and water of leaves, except for LP and C:P ratio according to Figure 4. Cerabolini (2010) revealed that the stress-tolerators tend to exhibit a greater lateral spread, with the competitors exhibiting only intermediate lateral spread according to the analysis of plant investing biomass of 506 species of 57 families for CSR classification; simultaneously, they also revealed the trade-off between rapid regenerative capacity versus investment in durable tissues [60]. In our results, component S had the highest contribution, while component R had the lowest contribution at the individual and community levels (Figure 2A,B), indicating S and R located two extremes in contributing typically to the leaf economy spectrum axis as the strategic trade-off of stress tolerance and reproductive regeneration. In the CSR framework, S and R strategies are the two ends of the typical plant economic spectrum, indicating that the S and R strategies correspond to the resource conservation and acquisition strategies, respectively [5,12,61], which strongly supports our result that the component S was significantly positively correlated with CWM− LC and C:N ratio representing carbon conservation, but with a significant negative correlation with CWM− LWC, LN and N:P ratio, meaning nutrient resource acquisition while displaying a converse tendency for component R (Figure 4). In addition, karst habitats generally lack nutrients, with lower nitrogen and extremely low phosphorus [4,58,62,63]. Therefore, karst plant communities may suffer nutrition-deficit limitations in survival. Fujita (2014) found that plants in phosphorus-limited habitats would invest less in sexual reproduction [64]. It supports our observation of the least R component contribution and the most S component contribution with the intermediate C component contribution. Synthetically, the stress tolerance strategy dominates karst forests depending on the trade-offs of competition and reproduction by the CSR analysis in association with plant functional traits, similarly verifying our hypothesis of H2 that CSR strategy depends on trade-offs in regulations associated with plant functional traits.

In addition, the Mantel's tests displayed that CSR components were significantly associated with plant functional traits in growth size and leaf stoichiometric traits and leaf water showing different trait combinations at individual and community levels (Figure 5), probably due to the variance from individual to community scale. Eichenberg (2014) suggested that these traits of SLA, LN, LDMC and LC were related to photosynthesis and protection and defense [65], and further as a trade-off explanation between rapid growth and survival by Chai (2015) [66]. For example, plants maintain high LDMC and LC for leaf structure protection to prevent water loss for photosynthesis as the substantial life defense [25], which strategically supported S selection in plant survival and investment suggested by Zhou (2021) [42], corresponding to our results of component S positively correlating with CWM− LC and C:N ratio and negatively with CWM−LN (Figure 4D,E,G) in strategic trade-offs associated with plant functional traits.

Moreover, previous studies have shown ecological strategy trade-offs due to the long-term evolution adaptivity through a series of environmental conditions [47,67]. The plant functional traits did not independently vary in integrated performance because any particular trait was constrained by its functional relation to other traits, according to Lavorel (2002) [68]. Novakovskiy (2016) analyzed the leaf functional traits and CSR components of

74 plant species from European mountains and plains; they found that species with stress tolerance strategy had higher LDMC, LC and lower SLA, LN with ruderal strategy exhibiting low LDMC and high SLA, indicating ecological strategy associated with leaf functional traits [69]. Similarly, Zhou (2021) also reached the same foundation of that ecological strategy depending on leaf trait variations in the degraded alpine meadow communities [42]. Therefore, these researches supported our results in community traits associated with CSR strategies, which supported our hypothesis H2 regarding CSR strategies of stress tolerance and competition and reproduction depending on trade-offs in regulations associated with plant functional traits.

*4.4. Strategy Mechanisms of Karst Forests*

The karst ecosystem generally presents a discontinuous soil cover, poor water retention, and deficit nutrients such as nitrogen and phosphorus [4,58,63], further influencing plant development and leaf trait variations [70]. Karst plants often confront water and nutrient shortages [58,71], signifying growth maintenance by the S strategy selection of stress tolerance under the harsh habitat of resource limitation, according to the CSR framework of Grime (1977), with developing high LDMC, leaf carbon and low leaf area, leaf nitrogen in adaptivity regulations [2]. For instance, Liu (2022) found that karst forests developed high LDMC and LC and low SLA and LN [25], revealing a strategy in nutrient acquisition and resource conservation relative to non-karst communities under the same climate zone [39–41]. Gerard (2008) discovered that high species diversity was related to CSR strategy with high-stress tolerance to a plant community [72]. The karst ecosystem maintains high biodiversity due to the high heterogeneity providing more niches for ecospecies despite suffering from severe resource constraints [35,73,74]. In our results, component S had the most contribution compared with the secondary C and the least R, regardless of individual or community levels, indicating that most karst plants suffer long-term stress tolerance. Rosenfield (2019) suggested that woody plants generally have a relatively lower R contribution, also supporting our results [52]. In addition, nutritional limitations provide little investment in sexual reproduction [64], aiming to maintain survival in stressful environments exactly as the most S and the least R in this study.

In strategy regulations, plants with high LA improved photosynthetic production incomes, but reduced defense inputs in the leaf-dried matter content of LDMC [42,69,75]. However, the high SLA combing with high LN and LWC could quickly harvest resource profits and rapidly promote reproduction [60,76]; namely, the R plant strategy invests resources in reproduction to compensate for the impact of harsh environments [27,69]. These researchers explained the strategy trade-off in regulating plant functional traits from adapting to different environments, which was consistent with the results of component R positively correlating with LWC and LN and negatively with LC and C:N ratio (Figure 4). Liu (2022) revealed that karst forests adopt a conservative survival strategy in viewing plant functional traits of the community, via low SLA and LN and high LDMC, LC and tissue density in decreased investments of leaf macronutrient N and P [25]. Here, the low SLA and LN and high LDMC and LC indicated the stress tolerance for karst plants, attributed to low productive maintenance and high leaf structural defense with weak reproduction. Notably, component R increased while component S decreased in an increasing CWM−N:P ratio (Figure 4I), indicating a strategic trade-off between R and S, which was inconsistent with the discovery of component R decreasing with the increased N:P documented by Fujita (2014) [64]. In our study, the mean N:P ratio was 20.32 (Table 1); Koerselman and Meuleman (1996) discovered that an N:P ratio > 16 indicated P limitation on a community level, while an N:P ratio < 14 indicated N limitation, as the co-limitation between 14 and 16 [77]. Therefore, karst vegetation always suffered P limitation. The increasing N:P promoted component R to rise, but simultaneously improved component C, resulting in the reduction of component S. Consequently, element P is probably the key factor in regulating karst community adaptation to maintaining survival and competition regulation and reproduction. Overall, through CSR insight, the predominant stress tolerance strategy trade-

off competition and reproduction dominate the karst forests, adaptively regulating plant functional traits. Simultaneously, the disentangled ecological strategies provide a theoretical perspective as an evolutional adaptive mechanism of karst vegetation, contributing to the significance of understanding mechanisms of community function maintenance.

## 5. Conclusions

In this study, our findings revealed that most woody individuals and communities in the karst forest tended to adopt strategy S (stress tolerance) with C (competition) based on the CSR component analysis, indicating that high-stress tolerance with competition predominates the karst forests. Variations in plant functional traits of leaf and plant size suggested trait differentiation and a significant correlation between CSR components and PCA scores. Moreover, our results demonstrated strategic trade-offs associated with plant functional traits by S relative to C and R, via oppositely trending regression of CSR components. Furthermore, Mantel's tests indicated that different CSR strategies had distinct key trait combinations. To summarize, the predominant ecological strategy of stress tolerance is based on the trade-off relative to competition and reproduction, in association with plant functional traits of plant size, leaf macronutrients, and leaf water. The disentangled strategy trade-off provides a theoretical point of view into karst forests, contributing to the importance of understanding community function maintenance as an evolutionary regulation mechanism of karst vegetation.

**Author Contributions:** Formal analysis, X.H., L.G. and K.S.; funding acquisition, Y.H.; investigation, Q.T., L.K. and Z.F.; methodology, Y.H.; supervision, X.H., M.U., Y.G., K.S. and T.X.; visualization, X.H.; writing—original draft, X.H.; writing—review and editing, Y.H. and M.U. All authors have read and agreed to the published version of the manuscript.

**Funding:** This work was supported by the National Natural Science Foundation of China (NSFC: 32260268), the Science and Technology Project of Guizhou Province [(2021) General-455], the Guizhou Hundred-level Innovative Talents Project [Qian-ke-he platform talents (2020) 6004], the Natural Science Project of Guizhou Minzu University [GZMUZK(2022)YB14].

**Data Availability Statement:** The original contributions presented in the study are included in the article, while further inquiries can be directed to the corresponding author.

**Conflicts of Interest:** The authors declare no conflict of interest.

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
