# Peer review of "Strategy Trade-Off of Predominant Stress Tolerance Relative to Competition and Reproduction Associated with Plant Functional Traits under Karst Forests"

_forests, doi:10.3390/f14061258_

Round 1

Reviewer 1 Report

The article advances the frontiers of knowledge by demonstrating that the karst plant community has evolved a stress tolerance strategy related to the economic spectrum of the leaves. These results are important to understand and propose alternatives for conservation of this ecosystem.

The article, needs to go through an adaptation process. Some points were raised in the methodology and others in the discussion of the work (pdf file). My biggest concern is with the denotation that the authors want to give to the term degraded karst forests. Because, if it is in relation to the natural stress conditions provided by the environment, this term can lead to confusion in readers. Another point of concern is the results at the individual level, but the authors are referring to the species level.

Author Response

Q1: Lines 54-55, please check if this sentence is complete.

Response: we revise this sentence in line 55.

Q2: Lines 93-94, verify if this hypothesis has already been tested in other works. Or specify that this hypothesis is for Karst forests.

Response: Your comment is very good. We proposed hypothesis 2 was based on the finding of Díaz (2023) that plant functional traits reflect the survival adaptation strategies adopted by plants to maximize carbon harvest and are a reflection of trade-offs in the allocation of resources such as plant structural composition and nutrient use. Araujo (2023) found that plant species have different ecological strategies and exhibit a diversity of trait combinations and trade-offs. Therefore, we believe that the ecological strategies of plants are strongly correlated with the functional traits of plants. So hypothesis 2 is actually specializing in karst forest ecological strategies. Given your suggestion, we revised hypothesis 2 to 'we hypothesized that CSR strategy depends on trade off in regulations associated with plant functional traits in karst forests (H2)', see lines 93-94.

Q3: lines 102-105, what is the source of this information?

Response: Thanks a lot for your comments. This information was provided by the local meteorological bureau.

Q4: Lines 105-108, what are the striking differences between the plots in terms of edophoclimatic conditions, successional stage and degree of environmental degradation?

Response: Many thanks for your good comments. The study areas we selected were forest vegetation controlled by the same climatic zone. There were no highly significant differences between these plots. In our study, we only focused on how the ecological strategies of woody plant communities developed under such degraded karst habitat conditions are associated with functional plant traits. Therefore, we did not consider the changes in CSR strategies during vegetation succession, but that must be a direction worth studying and exploring. In fact, we can often find that in the same climate region, there are large differences in community composition and structure in different stages of degraded karst vegetation succession (e.g. herbaceous community, shrubland community, tree forest community and top community stages) in the same climatic region. The forest vegetation involved in this paper represents the most typical and widely distributed vegetation type in southwest China, which was relatively well preserved, but they are still secondary forest vegetation developed by gradual succession of degraded habitats. In addition, we also believe that the recovery of forest vegetation due to environmental degradation, with significant differences in composition and structure at different stages of recovery according to the natural successional sequence, will be an interesting topic and will become the direction of attention in our next work.

Q5: Line 110, How did these inclusion criteria manage to include the layer of herbs?

Response: Thank you for your kind comments. Our manuscript studies the ecological strategies of karst woody plants. Therefore, herbal layer plants were not included in the investigation.

Q6: Line 126, So high temperature. Cornelissen suggest 60 - 80 C,

Response: Thank you for your comments. We revised the temperature to 65℃ in line 129.

Q7: Line 126, Was it for 48 hours or until constant mass?

Response: We are very grateful for your comments. We revised this sentence to “The leaves were rehydrated for 6-12 h before measuring the leaf dry weight (LDW, mg), then leaves were oven-dried at 65 °C for 48 h to constant weight and weighted as LDW” in order to reduce confusion, see lines 127-129.

Q8: Line 128, The authors should have used water-saturated leaf mass. Leaf rehydration is a fundamental procedure for determining LDMC, as suggested by Cornelissen.

Response: Many thanks for your good comments. For the determination of LDMC, we tracked the references of Cornelissen and provided additional descriptions for the determination of LDW, see lines 127-129.

Q9: Line 168, Check if you are referring to species or individuals.

Response: Thank you for your reminder. We conducted a comprehensive check of the data, and our research refer to the individual level, and we made corresponding modifications to the description in the text.

Q10: Line 180, What do the different colors mean?

Response: Thank you for your comments. The different colors in Figures 2A and B represent different individuals and communities, respectively.

Q11: Line 180, Do the dots represent species or individuals?

Response: Thank you for your comments. The dots in Figure 2A represent individuals, and we revised 'species' to ' individuals ' in the new version; the dots in Figure 2B represent communities.

Q12: Line 241, Species or individuals? It would be clear.

Response: Thank you for your comments. We conducted a comprehensive check of the data, and our research refer to the individual level, and we revised 'species' to ' individuals ' in the new version.

Q13: Lines 248-254, The authors built the hypothesis based on evidence from degraded environments. Degraded environments lead to S strategies. But now in the discussion they use the successional stage as an argument to explain the confirmation of the hypothesis. It got pretty confusing and disjointed.

Response: We are very grateful for your comments. Considering that our research did not involve succession, we have removed the sentence ' Meanwhile, the ecological strategies can shift each other in different succession stages, precisely as the differential trait patterns in different successional stages with variable key traits predicting species replacement during the succession process.' to make the discussion coherent.

Q14: Lines 289-293, This sentence is very confusing.

Response: Thank you for your comments. Stress-tolerators tend to exhibit a greater lateral spread, with competitors exhibiting only intermediate lateral spread because lateral spread is not necessarily a purely competitive ‘‘size’’ related trait involved solely in resource acquisition and foraging. Competitors are characterized by larger individual and extensive lateral spread both above and below ground; meanwhile, we found that component C was positively correlated with DBH, Height and LA. In addition, in the CSR framework of Grime (1977), species were ordinated in the C-dimension according to a 'competitive' index. This was a composite of canopy height and lateral spread. In the S-dimension, species were similarly ordinated according to the maximum relative growth rate in the seedling phase. Moreover, according to Cerabolini (2010), competitors represent a compromise at the center of this economics/ reproduction trade-off, because they must have high LA enough to capture light resource to allow rapid growth suggested by Wang (2023). In contrast, ruderals experienced little competition, leaves height was lower, and had strong reproductive ability.

Q15: Line 339, Authors need to be careful with the use of this term, as it suggests anthropic degradation. I don't think that's what the authors mean.

Response: Thank you for your comments. Ecosystem degradation is a natural process, along with the abduction and acceleration of human activities. Natural and human activities leading to forest degradation may lead to changes in plant ecological strategies. In our research, we only focused on the correlation between plant ecological strategies and plant functional traits, without considering the factor of habitat degradation. However, this is also a direction worth studying and exploring. Therefore, we have made corresponding modifications and deletions to the term 'degradation' in the full text, see blue font. Furthermore, we revised the title to 'Strategy trade-off of predominant stress tolerance relative to competition and reproduction associated with plant functional traits under karst forests' in this revised version, to better support the conclusion that is consistent with our results.

Reviewer 2 Report

The article was devoted to the ecological strategy for competition, stress tolerance and ruderal based on the CSR framework with functional features of leaves. The aim of learning about the ecological strategy was to understand the mechanisms of functional regulation in degraded forest ecosystems.

The field study was conducted in a typical karst forest located in Machang, Xiayun and Gaofeng Town, Pingba District, Anshun City, Guizhou Province, China.

A total of 53 plots with a total of 212 sub-plots measuring 10 m × 10 m were examined. Several indices were calculated: DBH, LA, LFW, LDW, SLA, LDMC, LWC, LC, LN and LP. The CWM value was used to assess the CSR strategy. Using the "StrateFy" spreadsheet provided by Pierce, the values ​​of the C, S and R components were calculated for each community-integrating species. Principal component analysis and the Mantel test were used for the analyses.

I propose to attach a map with the location of the research plots to the typescript.

Author Response

Q1: I propose to attach a map with the location of the research plots to the typescript.

Response: Thanks for your suggestion. We added Figure 2 regarding the location of the research plots, see Figure 1.

Other revision: We corrected some points according to Reviewer 1 and Reviewer 3. Mainly see blue font in new version. In addition, we revised the title to 'Strategy trade-off of predominant stress tolerance relative to competition and reproduction associated with plant functional traits under karst forests' in this revised version, to better support the conclusion that is consistent with our results.

Reviewer 3 Report

Detailed comments:

-          Pg.1, Keywords; Keywords should not match the title of the paper. I recommend omitting some keywords (those that are repeated with the title of the manuscript) or replacing them with others. They need to be changed.

-          Pg.5, Results; Ln 185 – ….LA (38.66%) and CV value in Table 1 (38.77%)? – what is correct?

-          Pg.5, Table 1, From a mathematical point of view, the standard deviation values should be indicated with a ± sign.

-          Pg.7, Figure 3, Explanatory notes are missing under the Figure R2 and P. Moreover,  R2 = coefficient of determination?  

-          Pg.8, Ln 246 - ….. Caccianigasuggested…..it must be divided.

Author Response

Q1: Pg.1, Keywords; Keywords should not match the title of the paper. I recommend omitting some keywords (those that are repeated with the title of the manuscript) or replacing them with others. They need to be changed.

Response: Thank for your suggestion. We changed “CSR strategy” to “CSR theory”. We added the keyword “ecological strategy”, “plant size”, leaf macronutrients”, and deleted “plant functional traits”, “degrade forests”, see lines 31-32.

Q2: Pg.5, Results; Ln 185 – ….LA (38.66%) and CV value in Table 1 (38.77%)? – what is correct?

Response: We checked and recalculated the data that the CV of LA Table 1 is right, and we revised “38.66%” to “38.77% ” in line 189.

Q3: Pg.5, Table 1, From a mathematical point of view, the standard deviation values should be indicated with a ± sign.

Response: Thank you for your a lot comments. We revised it to “Mean±SD, see Table 1.

Q4: Pg.7, Figure 3, Explanatory notes are missing under the Figure R2 and P. Moreover,  R2 = coefficient of determination?  

Response: We are very grateful for your comments. We added the explanatory notes of R2 and P in lines 225-226.

Q5: Pg.8, Ln 246 - ….. Caccianigasuggested…..it must be divided.

Response: Thank you for your reminder. We divided it in line 250.

Other revision: We corrected some points according to Reviewer 1 and Reviewer 2. Mainly see blue font in new version. In addition, we revised the title to 'Strategy trade-off of predominant stress tolerance relative to competition and reproduction associated with plant functional traits under karst forests' in this revised version, to better support the conclusion that is consistent with our results.
